# Training Community Pharmacy Staff How to Help Manage Urgent Mental Health Crises

**DOI:** 10.3390/pharmacy7030133

**Published:** 2019-09-16

**Authors:** Nathaniel Rickles, Albert Wertheimer, Yifan Huang

**Affiliations:** 1Department of Pharmacy Practice, University of Connecticut, Storrs, CT 06269, USA; yifan.huang@uconn.edu; 2Department of Sociobehavioral and Administrative Pharmacy, Nova Southeastern University, Fort Lauderdale-Davie, FL 33314, USA; awertheime@nova.edu

**Keywords:** mental health care, pharmacy staff, mental health first aid, mental illness, pharmacist roles

## Abstract

Nearly 44 million Americans are affected by mental illness every year. Many individuals, however, are not diagnosed and/or do not receive treatment. The present manuscript reviews the incidence of mental illness, the continuum from mental wellness to mental illness, and the role of the pharmacy staff in helping individuals manage different mental health needs. In particular, there is discussion of stigma of mental illness that those with mental health needs experience by those around them including health professionals such as pharmacy staff. One way to resolve such stigma is through training such as Mental Health First Aid (MHFA). The paper reviews key aspects of MHFA, the evidence supporting MHFA, and how MHFA relates specifically to pharmacy practice and services. A conceptual framework for MHFA and its relationship to individual factors, attitudes, behaviors, and outcomes. Lastly, a discussion is presented that briefly compares MHFA to other similar approaches to helping those in mental health crises, the limits of what is known about MHFA, and what future research might explore to better understand the outcomes of pharmacy staff providing mental health education, support, and referral to care.

## 1. Background

There are at least 43.8 million individuals in the US affected by mental illness every year [1]. There are several more millions of individuals, undertreated, who would be diagnosed and properly treated for mental illnesses who are not in the system [2]. In addition, there are numerous other individuals who do not meet the criteria for a diagnosis of mental illness or require treatment but experience episodes where they are unable to optimally cope or manage their temporary thoughts, feelings, and behaviors. For a variety of reasons, whether for those already diagnosed with mental illness, needing to be diagnosed and/or treated, or those with more transient/temporary mental health symptoms, these groups of individuals face significant challenges in obtaining resources to help them initiate and/or remain consistently engaged with a support system that can provide the needed support to help them resolve their temporary or long- term difficulties. Key reasons why these groups of individuals struggle to initially reach out for support or consistently engage in support include (1) the stigma experienced by all these groups that are afflicted with different levels of mental health challenges face, and (2) the lack of access to individuals trained to provide temporary supportive management of specific mental health crises and know when and what types of additional support is needed for such individuals [3].

The National Alliance on Mental Illness (NAMI) and many other mental health advocacy groups and organizations have promoted anti-stigma campaigns to target the first of these two reasons [4]. Clearly, there is still much work to do to make individuals with a variety of mental health symptoms feel more comfortable reaching out and having someone help them through their symptoms. There are many efforts that continue to focus on facilitating cultures at work places, health care environments, and other areas of the community that avoid judgement and distance away from the individual suffering from ongoing or temporary mental health symptoms [5]. One common example is using “person first” language that avoids labeling individuals by their illness such as “schizophrenics” and rather referring to these individuals as those with lived experience of schizophrenia or “individuals with schizophrenia.” Rather, the focus of this anti-stigma work is to help bring the affected individual together with a community resource and together support the individual towards improved mental health management. As valued, trusted members of the community, pharmacists are at the frontlines of primary health care and thus regularly interact with individuals who may have mental illness and mental health crises. If pharmacists were equipped with tools that enhanced their engagement of individuals with mental health needs, it is possible pharmacists could help adults in need of mental health services seek mental health care. Mental illnesses are so prevalent throughout society, yet they are also some of the most severely undertreated. While 1 in 5 adults in the US is afflicted by mental health conditions in a given year, only 41% of those adults received mental health services [1]. Among adults with a serious mental illness, 62.9% received mental health services in the past [1]. Additionally, just over half (50.6%) of children with a mental health condition aged 8–15 received mental health services in the previous year [1]. Such disparities contribute to significant costs to the individuals with mental health symptoms and the healthcare system as a whole. Not only are individuals unable to optimally function that affects their quality of life, but they are actually also at greater risk for many comorbid conditions. On average, adults in the US living with mental illness die 25 years earlier than others, largely due to treatable medical conditions [1]. Suicide is also the 10th leading cause of death in the US, the 3rd leading cause of death for people aged 10–14, and the 2nd leading cause of death for people aged 15–24 [1]. Not only does mental illnesses severely impact individual lives, it has also contributed to $193.2 billion in economic losses per year [1].

Identifying mental health concerns and illness can be thought of on a continuum from what is considered mental wellness to mental illness (Table 1). The model presented in Table 1 is based on the definition and concepts in the Mental Health Continuum Model [6]. On the mental wellness end of the spectrum, we would expect healthy reactions including predictable mood fluctuations, general acceptance of challenges, sense of humor, high performance, average sleep patterns, moderate to high level of energy, and avoidance of excessive use of substances and engagement in addictive behaviors. Moving to the middle of continuum, an observer might start noticing the following: more unpredictable mood fluctuations, difficulty managing life challenges, diminishing sense of humor, reduced performance, poor sleep patterns, and moderate to low energy, more anxiousness about different activities, and gradual engagement in unhealthy behavior patterns. At the ill end of the spectrum, an individual will be observed to have: angry outbursts that are more significant and frequent, very labile moods with very low and/or low moods, significant anxiety, presence of unusual thoughts and behaviors, poor sleep, significant difficulties performing in daily activities and work, excessive substance use and risky and/or addictive behaviors, and thoughts of harming the self and others. Individuals presenting at different stages within this continuum are in need of different levels of support and professional help. Thus, awareness of the mental health continuum acts as a useful resource and context to clarify the severity of impairment of those suffering from mental health crises and illnesses, and what resources such individuals may need. Although Table 1 depicts a list of signs and symptoms under each aspect of the continuum from mental wellness to mental illness, this should not be interpreted that an individual needs to have all the signs and symptoms listed at one time to be considered mentally well or mentally ill. An individual may present with signs and symptoms from different parts of the continuum. 

While poor mental health can lead to poor physical health and mental illness, it is possible to have poor mental health temporarily and not lead to a diagnosed mental illness. For example, a person going through a hard life experience may have poor mental health during this time period without necessarily having a mental illness. Thus, feeling miserable or isolated may be red flags that your mental health requires attention and improvement. In these cases, it is important to utilize support from friends/families/other groups, focus on healthy lifestyle habits such as eating regularly and appropriate sleep hygiene, and even seeking support from a therapist can be beneficial. Poor mental health not leading to mental illness can be due to difficult life transitions, environmental challenges, and the lack of needed resources. It is also important to note that poor mental health may not lead to poor health and that an individual with mental illness may be in good physical health.

Diagnosed mental illnesses, however, are often associated with physiological alterations in the brain due to genetic predisposition, chemical changes, traumatic experiences, and/or major developmental challenges leading to significant changes in mood, thinking, and behavior that cannot be easily managed with self-help strategies or non-professional support systems. These individuals often require referrals to their primary care doctor, a psychologist, a psychiatrist, and occasionally a social worker for management of the various aspects of treatment. Some more severe cases may require hospitalization, or treatment programs with more intensive interventions and monitoring.

## 2. Role of the Pharmacy Staff

Pharmacists are considered the most accessible health care professionals, and see a large volume of patients in community pharmacies every day. The College of Psychiatric and Neurological Pharmacy (CPNP) and National Alliance on Mental Illness (NAMI) collaborated on a 2012 national survey and assessed the opinions of 1031 individuals with mental health concerns, or their caregivers, regarding their relationship with their pharmacist. Ninety-one percent felt very comfortable going to their community pharmacy, and 83% felt like the pharmacist respected them [7].

Pharmacists are involved in many services including: counseling, monitoring, and reducing side effects, assessing serum concentrations, identifying interactions, assisting in treatment plans, and developing ways to increase medication adherence. Pharmacists have also been repeatedly shown to enhance patients’ knowledge, beliefs, and sense of treatment progress related to psychotropics such as antidepressants [8]. These findings illustrate the impact that pharmacists can have on the treatment, attitudes, and perceptions of their patients with mental illnesses. Such impact can help improve patient medication adherence. Pharmacy staff already have access to a large segment of this population. If pharmacy staff have the appropriate mental health skills, they can help to reshape the negative perceptions individuals with mental health needs and illness have about themselves and their illnesses. Such pharmacy support can remove a significant barrier to care, to be discussed in greater detail in next section [9]. In addition, pharmacy staff knowledgeable in the signs and symptoms of mental illnesses can also help identify individuals that may need to be referred to a clinician for follow-up and support. Pharmacy staff also can be excellent resources of information about mental health conditions and encourage various self-help and other strategies.

## 3. Barriers to Care: Presence of Stigma

A significant barrier to care for those individuals with mental illness is the stigmatization of their illness by others. Stigma has been developed and supported by a combination of media incorrectly portraying mental illness, and a lack of training/education regarding mental health awareness. With movies like *Friday the 13th—A New Beginning* (1985), *Psycho III* (1986), *Misery* (1990), and *Silence of the Lambs* (1991), many people have grown up with the misconceptions that mental illness disenables a person from logical thinking and makes them difficult to reason with, and even dangerous [10].

Stigma persists throughout all areas of the community, and healthcare is no exception [9,11]. In general, some healthcare professionals may view individuals with mental illness as incompetent, unpredictable, dangerous, difficult to communicate with, and require too much time to provide service. Even insurance companies have been shown to offer greater coverage for physical illness over mental illness [12]. Studies generally show that pharmacists have a positive view of people with mental illness, but are often uncomfortable with them [9,13,14,15,16]. Upon survey, pharmacists reported being more willing to counsel on a physical disorder (i.e., asthma) than a mental disorder [16].

In the 2012 national survey by CPNP and NAMI mentioned previously, a little over half of those sampled reported having a strong professional relationship with the pharmacist [7]. Approximately 40% of those in this latter sample actually reported no relationship with their pharmacist and 75% reported not receiving effectiveness and/or safety monitoring assistance from the pharmacist [7]. This illustrates a need for greater education focusing on how pharmacy staff can feel more comfortable in communicating with individuals with mental health needs and illnesses. Such greater comfort in communication will foster a stronger relationship between individuals with mental health needs, mental illnesses, and pharmacy staff. Through these stronger relationships, pharmacy staff can provide much needed education and motivational resource for individuals with mental health needs that may otherwise be overlooked by their system of care. Pharmacists need more training and skills to help improve their comfort in communication and build relationships with individuals with mental health needs. Such mental health training opportunities will be discussed in the next section.

## 4. Mental Health First Aid

Mental Health First Aid (MHFA) is a training program offered to the general public aiming to increase mental health literacy, and impart the participant with the skills required to provide an immediate response to a person experiencing an acute mental health crisis [17]. MHFA also helps train the participant to understand the signs and symptoms of different mental illnesses so they can help someone seek the appropriate support especially in the earlier stages of developing a mental health disorder or problem. The program originated in Australia in 2001 and was designed by Kitchener, a nurse specializing in health education, and Jorm, a mental health literacy professor. The program can be given as a one 8-hour day, two 4-hour days, or four 2-hour days. Programs delivered over the course of multiple days should not occur more than two weeks a part. During this time, participants obtain the skill set necessary to not only intervene in a mental health crisis, but to identify those at risk and/or having symptoms of mental illness, and direct them to the help they need. In both the participant and MHFA instructor manuals, MHFA is presented as being like first aid needed for cardiovascular support (American Red Cross’ CPR program) but for mental health crises.

The program focuses on depression and other mood disorders, anxiety disorders, trauma, psychosis, and substance use disorders. Participants in the training learn the program’s “Mental Health First Aid Action Plan,” and are taught to apply the action plan to many different scenarios—specifically guiding interactions with others experiencing suicidal thoughts or behaviors, panic attacks, non-suicidal self-injury, overdose or withdrawal from substance use, and reaction to a traumatic event. The action plan consists of five steps:Assess for risk of suicide or harmListen non-judgmentallyGive reassurance and informationEncourage appropriate professional helpEncourage self-help and other support strategies

The first letter of each step forms the acronym ALGEE. Throughout the training, instructors frequently refer to ALGEE and reinforce the ALGEE action plan.

There are no requirements for the training; it is available to the general public. The National Council for Behavioral Health (NCBH) that accredits the training programs strongly encourages professionals to participate who have frequent contact with those at risk or experiencing mental illness/crisis. Specific examples listed on the MHFA website include police officers, human resources workers, primary health care workers, schools, community faith organizations, and friends and family of someone experiencing mental illness or mental health crises. The Community Pharmacy Foundation supported a project by the National Community Pharmacist Association (NCPA) to develop and provide 8-hour of continuing education credits for pharmacists and technicians who participate in the 8-hour MHFA courses. This project also supported the creation of specific pharmacy-related cases that MHFA instructors can use when presenting topharmacists and pharmacy technicians. These cases are available to certified MHFA instructors who complete several days of NCBH training for certification. To use these NCPA cases, there is no additional training beyond needing to be certified to train others in MHFA.

There is a small but growing literature on MHFA in general and its application to pharmacy. A meta-analysis was performed estimating the effects of the MHFA program in participants of any occupation, both for adults and young people, based on results published up to March 2014 [18]. MHFA was found to be effective in increasing knowledge regarding mental health problems, and effectively decreasing negative attitudes toward individuals suffering from mental health problems. The program was also shown to increase help-providing behavior. Another publication reviewed three published trials, and found improved concordance with health professionals about treatments, improved helping behavior, greater confidence in providing help to others, and decreased social distance from people with mental disorders [19].

In Sweden, researchers conducted a randomized controlled trial involving a group of staff from various agencies who randomly received MHFA training and those who did not [20]. Results showed that the MHFA group had significantly greater knowledge and confidence in helping others with mental health disorders than the control group [20]. Such MHFA effects were shown to be maintained to a great extent after a two-year follow-up period [20]. The collective data reviewed in this section reinforces the central finding that taking a course in MHFA can significantly improve a participant’s knowledge of mental health and mental health disorders, confidence in helping an individual manage a mental health crisis, decrease stigma, and sets the participant up for a more successful interaction when reaching out to those with mental illnesses.

In addition to the general literature regarding MHFA’s impact on trainee outcomes, there is specific growing literature on MHFA’s impact on pharmacists and pharmacy students. One study took place in eight rural Australian community pharmacies and entailed a survey assessing barriers to MHFA training and applications of the training to pharmacy [21]. Pharmacists identified and supported the need for MHFA, and had low confidence in their ability to handle an acute mental health crisis without the additional training [21]. The majority (72%) of pharmacists agreed they have a role to provide MHFA, but less than half (48%) were comfortable in providing this support. [21]. Most participants stated that they would be prepared to undertake this training. About a third of the sample thought the training should be administered during their intern year, 24% thought training should be administered during their pharmacist years, and the majority (76%) preferred administration of training as a one-day course with an online component [21]. The major barriers identified by pharmacists included time, geographic location, and resources [21]. These results show that pharmacists have both identified the need for and are willing to undertake MHFA training in order to improve access to mental health support in individuals at risk for or experiencing mental health crises and illnesses.

A controlled trial was conducted assessing 60 third year pharmacy students at the University of Sydney randomly chosen to complete one of two 12-hour sessions of MHFA and complete a follow up survey [22]. The survey (administered before and after the training) evaluated mental health literacy, the 7-item social distance scale, and 16 items related to self-reported behavior. Survey results showed that MHFA training reduced the pharmacy students’ stigma towards individuals with mental illness, improved recognition of mental disorders and improved confidence in providing services to individuals with mental illnesses in a pharmacy setting [22].

Another randomized controlled study of 262 members of the Australian public demonstrated that students (pharmacy and other programs) who participated in an online MHFA course responded better to measures of stigma reduction compared to those that used a written manual [23]. This indicates that completing a training program such as the MHFA is more valuable than just offering educational materials to students. While pharmacy programs often provide clinical background on mental illnesses and evidence-based treatments to target these illnesses, they are typically less focused on helping students manage crises that might occur in their pharmacies. An additional study, taking place at University of Sydney, evaluated the differences in confidence and attitudes of their pharmacy students towards suicidal crisis in three groups of students: the first group simply completed MHFA training, the second group completed MHFA training and observed a simulation of a patient and pharmacist interaction with a person undergoing suicidal crisis, and the third completed the MHFA training and participated in the live simulated interaction [24]. Results displayed a statistically significant increase in confidence in all students who participated in the MHFA training. In particular, the greatest increases in confidence were among pharmacy students participating in the live interaction. Such data suggests that participating in a live simulation is critical to the MHFA training experience. This combined literature shows the clear value and impact of MHFA on learner attitudes and behaviors toward those at risk for mental health crises and/or experiencing mental illness.

These findings are likely supported by an underlying conceptual framework for how MHFA relates to individual factors, attitudes, behaviors, and outcomes. The authors depict this conceptual framework in Figure 1 as it specifically relates to MHFA by pharmacy staff and its effects on both the pharmacy staff and the outcomes of individuals experiencing mental health concerns and illnesses. Moving from left to right, the framework starts with the concept that background factors of both the individual with lived experience of mental health concerns and illness and the pharmacy staff affect the extent to which stigma of mental illness exists for both the individual and the pharmacy staff. From past extensive research on stigma of mental illness, it is well known that culture, education on mental illness, personal contact with individuals with mental health needs and illness, and demographics all impact the extent to which stigma towards mental illness develops [9,16,25,26]. In addition, there is a robust literature and framework development supporting the role of these background factors on pharmacy staff perceptions of counseling, relationships with patients, and the willingness to provide care and related counseling services to those with mental illnesses [16,27].

In the middle of Figure 1, MHFA can be seen as a variable with bi-directional impact on (1) stigma involving pharmacy staff and that of individuals with mental health needs, (2) attitudes of individuals with mental health needs and pharmacy staff towards each other, and (3) services provided by pharmacy staff to those with mental health needs. The research reviewed in this section supports MHFA’s direct impact on these aspects of pharmacy practice [21,22]. Although not examined specifically, it is likely that the stigma changes and practice changes from MHFA would likely change the willingness to adopt and implement MHFA (reflected by the arrows going from stigma back to MHFA and practice-related perceptions and services back to MHFA).

The arrows from pharmacy stigma, stigma of those with mental health needs, practice-related perceptions and services, and outcomes are all supported by decades of stigma and pharmacy practice research on these topics [25,27]. Many of the non-pharmacy literature reviewed also support how MHFA changes knowledge, willingness to help, negative attitudes toward those with mental health needs, and other process outcomes [18,19,20].

As for the right part of the Figure 1 linking MHFA to outcomes, there needs to be more research demonstrating the links between MHFA and various patient and health system outcomes such as mental and physical health of individuals helped by MHFA, cost of care and improved treatment [28]. Feedback models, such as Figure 1, often suggest that outcomes or outputs of the model will affect model inputs such as background characteristics. If MHFA improves mental and physical health, it is expected that one’s background and experience would also change as well, and subsequently other relationships in the conceptual framework. One suggested research approach to testing the relationships in Figure 1 is through the secret shopper methodology whereby trained actors go into pharmacies showing signs of being in a mental health crisis and see how those trained in MHFA react differently from those not trained in MHFA and how the actor felt differently with a trained and untrained MHFA person. It would be useful to also conduct randomized controlled trials to see if practicing pharmacists trained in MHFA respond differently to individuals with mental health needs and illnesses than those not trained in MHFA. 

## 5. Additional Considerations

MHFA is designed for administration to the general public. As noted by Chowdhary and colleagues, it may be prudent to give healthcare professionals a different program tailored to their education, training and experiences [28]. These programs should also highlight that pharmacists are not being trained in psychotherapy and other treatments. Rather, pharmacists being trained in MHFA need to remove their clinical intervention hats and focus on the ALGEE action plan to manage a mental health crisis.

MHFA is not the only program available to train participants on how to deal with mental health crisis and communicate effectively with those suffering. Other programs include options such as Psychological First Aid (PFA) and Emotional CPR (eCPR). PFA usually focuses more on protocol following a disaster. MHFA is more general than PFA by focusing on any mental health crisis [29]. eCPR, however, is another training geared toward the general public to assist a person experiencing an emotional crisis using three components [30].
C = Connecting with Compassion and Concern to CommunicateP = emPowerment to experience Passion, Purpose and PlanningR = Revitalize through Re-establishing Relationships, Routines and Rhythms in the community

eCPR is believed to involve consumers with lived experience of mental illness more actively in the trainings and, therefore, the voices of those with mental illnesses are heard first hand throughout the trainings. With no head to head studies comparing the different available training methods, it remains uncertain that MHFA is superior to these other programs in preparing the general public and pharmacy staff for these crisis-based interventions. MHFA, however, is gaining significant momentum throughout the country and recently Walgreens has announced its plan to train their pharmacists in MHFA [31]. Pharmacists are becoming certified train-the-trainers and offering MHFA sessions to pharmacy students and practicing pharmacists.

## 6. Conclusions

Although there is not yet enough evidence to support that MHFA improves patient outcomes, there is considerable support that it decreases existing stigma regarding mental illness (a leading barrier to individuals seeking care and pharmacy staff providing care). There is also evidence that MHFA increases pharmacist and pharmacy student confidence in identifying, approaching, interacting with an individual who is undergoing a mental health crisis, and helping them seek appropriate care. These are significant steps in improving access to needed help for individuals with mental illness or experiencing a mental health crisis. However, with no head to head studies comparing MHFA to other programs, it is still unclear which of the available programs are superior in creating sustainable changes in pharmacy staff confidence and engagement of individuals with mental health needs.

With the pharmacist being one of the most accessible and most frequently visited healthcare professionals, community pharmacists are well positioned to bridge the gap between those with a mental health crisis and/or illness and access to the treatment and support they need to begin the healing process. Community pharmacists clearly need to augment their knowledge regarding how to identify and respond to a mental health crisis. Training programs, like MHFA, are well designed to reduce stigma of mental illness, increase knowledge and confidence on how to manage mental health crises and support mental illnesses. By having more community pharmacists trained in MHFA or other similar programs, community pharmacists can meet a critical public health problem by being able to identify the urgent needs of someone in a mental health crisis, offer non- judgmental support, and provide critical information that allows the individual with mental health needs to quickly access the appropriate resources to help the individual achieve greater mental and physical wellness.

## Figures and Tables

**Figure 1 pharmacy-07-00133-f001:**
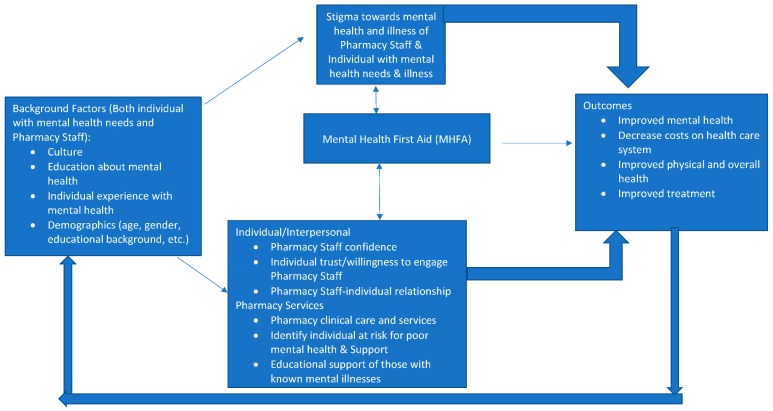
Conceptual framework of relationships between background factors, Mental Health First Aid (MHFA), and outcomes.

**Table 1 pharmacy-07-00133-t001:** Continuum of Mental Wellness and Illness.

Mental Wellness	Mild/Moderate Coping/Adjustment	Mental Illness Difficulties
Predictable mood fluctuations	Unpredictable mood fluctuations	Frequent & significant outbursts
General acceptance of challenges	Difficulty managing life challenges	Labile moods
Sense of humor	Diminishing sense of humor	Significant anger and/or anxiety
High performance	Reduced performance	Unusual thoughts and behaviors
Average sleep patterns	Irregular sleep patterns	Consistently poor sleep patterns
Moderate to high level of energy	Moderate to low energy	Significant dysfunction in daily
Avoidance of excessive use of substances	Anxiousness about activities	activities and work
Avoidance of addictive behaviors	Gradual engagement in unhealthy behavior patterns	Excessive substance use &/or addictive behaviors
		Thoughts of harm to self/others

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
