# Peer review of "Training Community Pharmacy Staff How to Help Manage Urgent Mental Health Crises"

_pharmacy, 2019, doi:10.3390/pharmacy7030133_

Round 1

Reviewer 1 Report

Dear author, while the topic is great, you will need to improve your approach to critical review.

Author Response

We did not see any specific comments from this reviewer except a general comment that we needed to improve our approach to critical review.  It is hard to respond to this comment without specifics.  

Reviewer 2 Report

Thank you for the opportunity to review your work. I think this paper was very well done. It builds on previous work done with MHFA and pharmacy. It is timely, and it (MHFA) is a topic/concept  needs more rigorous scientific evidence-based research to determine if client outcomes can be directly linked to MHFA trainings. Pharmacy staff are in an optimal position to provide MHFA care.

I think it is a fine paper in terms of a review of the literature or a preliminary concept paper- I would suggest that the authors provide some ideas/methods for testing their concept in pharmacypractice- what would that look like? I hope this helps. Comments: Page 1 line 22- substitute “would” for “should” Page 2 line 59-  no need for a separate paragraph beginning with “Suicide…” Page 3 line 98- start sentence with “Ninety-eight” spelled out Page 3 line 12- Omit “ Another huge” and begin with  something like “A significant barrier…” Page 4 line 149- omit the word “simple” Page 6 line 233- If there is a robust literature available, it would be best to see more relevant references Page 9 line 19- I would suggest moving the other mental health trainings that are available up higher in the paper- perhaps after the introduction of MHFA

Author Response

Thank you for the kind comments on the paper and we are glad this reviewer was positive about the manuscript and its value.

Per your reviewer’s suggestions:

We added a few comments in the discussion about how we might test the model in practice. A great suggestion. Thank you! We made the verb change from “should” to “would” on Page 1, line 22. Page 2 line 59: We removed the separation of the paragraph beginning with “Suicide. . .” It is now incorporated into the previous paragraph. Page 3, line 98: We spelled the number starting the sentence- the reviewer indicated “Ninety-eight” but meant “Ninety-one” as it should be. Page 3, line 12: We omitted the word “Another huge” and replaced with “A significant barrier” as suggested. Page 4, line 149: We removed the word “simple”. Page 6, line 233: The reviewer was asking about providing “more relevant references” for the robust literature the role of various background factors on pharmacy staff perceptions of counseling, relationships with patients, and the willingness to provide care and related counseling services to those with mental Without more information, it is hard to know what this reviewer would feel more relevant to the references provided further.  Further, we feel this extensive literature is well beyond the scope of the paper.  We feel a few references is sufficient to illustrate the point made and shouldn’t require more extensive review.  To also note, one of the two references provided is already an extensive review of the robust literature.  Page 9, line 19: While we appreciate the reviewer’s suggestion to move the presentation of psychological first aid and eCPR to the introduction of MHFA, we were unsure it would be more confusing there since it is hard for readers that early in the manuscript to appreciate how these latter trainings are different from MHFA. We thought it might be best to present MHFA and then bring forward briefly to the reader how these trainings are different from MHFA after seeing what is known about MHFA. We also think by ending the paper with comparison to these other trainings, readers will be left with the notion that more work is needed to compare MHFA results with these trainings.  If presented too early, such message may get lost by the time conclusions are made.  We will revise if reviewer remains concerned if not moved earlier in the paper.

Reviewer 3 Report

Thank you for the opportunity to review your article titled, “Training Community Pharmacy Staff How to Help Manage Urgent Mental Health Crises”.  It is a very timely and important topic. 

Please see below comments/suggestions:

A lot of the background as written sounds like commentary & is casual.  Consider rewriting to reflect more evidence/support from literature/citations in background. 

Line 43

Which mental health organizations?  Is the example regarding NAMI?

Line 48-49

There is no citation.  Where are you getting the number “millions” from?  Not sure you can predict how many more people would get mental healthcare.  I would consider revising the sentence.

Lines 59-61

This is very short to be a paragraph by itself.  Consider incorporating into the previous paragraph (lines 50-58)

Lines 62-77

I would consider creating a figure or table to show the continuum from mental wellness to illness. 

What is the source for the definitions of mental wellness and mental illness listed?  There is no citation for the paragraph.  Although someone can be well in terms of their mental health, that does not necessarily mean they are also well in terms of their physical health. Also, it appears you are stating as they move to mental illness they would have all of the signs/symptoms you have listed (“…an individual will be observed to have: angry outbursts that are more significant and frequent, very labile moods with very low and/or low moods, significant anxiety, presence of unusual thoughts and behaviors, poor physical health, poor sleep, significant difficulties performing daily activities and work, excessive substance use and risky and/or addictive behaviors, and thoughts of harming self and others”.  I would consider revising this paragraph and also adding in citation.

 Line 105-107

For the sentence, “This can be invaluable in the treatment of individual with mental illness, as many report stopping their treatment due to attitudes regarding their disorders, or the medication they are taking”.  Where is this from?  Consider including citation.

Line 118-119

The sentence, “Some healthcare professionals view individuals with mental illness as incompetent, unpredictable, dangerous, difficult to communicate with, and require too much time to provide service”.  Which healthcare professionals?  All or specific?  Consider including citation 

Line 135-137

In the first sentence of the paragraph, you are missing that not only does MHFA help with mental health literacy, stigma and someone in crisis, it also helps with the following: trains the participant to understand the S/S of different mental illness or diagnoses for identification, especially in the early stages of when a person may be developing a mental health problem.  I would consider adding something about that there (not just crisis).

Line 139

For the sentence, “The program occurs over eight hours but is often split into two four-hour sessions”.  Where is this from?  According to the instructor manual for MHFA, it can be given as one 8-hour day or two 4 hour days or four 2 hour days, not more than two weeks apart.  Did you find a reference where two 4 hour days were preferred?  Consider revising.

Line 142

Who is “Many” in the sentence “Many consider MHFA analogous to the first aid needed for cardiovascular support (American Red Cross CPR program) but for mental health crises”.  Please revise.

Line 160 & Line 165

Not sure want to include the links in the text.  Links can change over time and may be deleted in the future.  Also, may appear as if promoting NCPA.

Make sure to include limitations of training (ex: does not train you to be a therapist, etc) in the article

Line 201

“…chosen to complete two 12 hour sessions of MHFA…”  As written, it sounds like you are stating they completed two 12 hour sessions.  Please review study and revise if that is the case or were they assigned to 1 of 2 12 hour sessions?  Usually MHFA limits a class size to 30 participants per instructor (unless special permission given previously).  Also, I believe it used to be a 12 hour course that has now been decreased to a 8 hour course. 

Lines 222-223

Why is this sentence its own paragraph?

Figure 1: Consider revising title to include “conceptual framework” or to make more obvious concept since outcomes are not based on current evidence

Author Response

Reviewer suggests background seems like commentary and to consider rewriting section as more evidence/support from literature/citations. We appreciate this suggestion and added some references to support statements.  With that said, we feel much of the background reflects common knowledge and experience.  The background is really context for the focus on mental health first aid in pharmacy practice.  Line 43: (assume reviewer meant line 38 of reformatted manuscript by MDPI).  Reviewer was asking about which mental health organizations were being referred to in the sentence.  NAMI could be an example but the statement is more general about various ongoing efforts available to reduce stigma in multiple environments.  Reworked the sentence so that it is more inclusive and less specific to mental health organizations. Line 48: We revised the sentence as suggested. Lines 59-61: We incorporated this small paragraph into the previous paragraph as suggested. Lines 62-77: We created a more Figure 1 regarding the continuum from mental wellness to illness and have added it to the paper. The reviewer asked for citations to support definitions in paragraph.  We added a citation.  We also revised the paragraph to remove physical health in the definitions.  We also revised the text following the discussion of the Figure/model to clarify about the relationship between poor physical health and poor mental health and that being at any point in the continuum doesn’t mean one has to have all the signs and symptoms listed at that point in the curriculum.  We also added a point that an individual may present with signs and symptoms from different parts of the continuum. We thank the reviewer for these important suggestions. Lines 105-107: We removed the sentence “This can be invaluable in the treatment of individual with mental illness. . .” We replaced with a more simple sentence that does not require a citation. Line 118-119: The reviewer was asked for specifics on the statement that “some healthcare professionals view individuals with mental illness as noncompetent . . . “This was intended to be a general statement about healthcare professionals with stigma and not specific to a particular group of professionals. We added language to suggest the general nature of the comment and do not believe the statement requires a citation. Line 135-137: We added a comment about how MHFA assists with understanding the signs and symptoms of different mental illnesses for identification and especially early in stages when they are developing a mental health problem. Thank you for the suggested language. Line 139: We revised the wording of how the MHFA is delivered to match what was suggested by the reviewer (we confirmed with information in the MHFA instructor portal). We thank the reviewer for ensuring more precise language. Line 142: We removed the word “Many” from the sentence that “Many consider MHFA analogous to the first aid . ..” We slightly revised the sentence as well. Line 160 & Line 165: We removed links. We revised the text about NCPA and do not feel it is promotional in nature and felt okay to leave as is. We already included the limitation that MHFA does not train individuals in therapy, etc.  Please see section on additional considerations. Line 201: Reviewer asked that we clarify if O’Reilly and colleagues (2011) were randomly assigned to two 12-hour MHFA sessions or one of two 12-hour sessions of MHFA. We clarified in the text that it was random assignment of one of two 12-hour sessions of MHFA. Line 222-223: Not sure why the sentence was formatted as its own paragraph. We fixed it and incorporated with paragraph following it. Figure 1: We revised title of Figure 2 to include “Conceptual Framework.”

Round 2

Reviewer 1 Report

Dear author thank you for making the corrections

Author Response

Thank you for the positive feedback and see no further changes are needed.

Reviewer 3 Report

Overall great job.  The literature search/information you added to the article since the last submission significantly contributed to the paper.

The following are some suggested comments/edits:

Line 8:  What do you mean about "properly" treated?

Line 12:  Recommend to delete "the" between through and training.

Line 15:  Need space between end of sentence and "Lastly".

Line 32:  Not sure "convenient" is a good term to describe.  Was that how it was described in the reference?

Line 63:  This is where first reference is to "Figure 1", but I did not find it in my draft copy.  Please make sure included in the submission.

Line 99-115:  You describe the role of the pharmacy staff, but it is all about stigma.  Under this section, I would consider elaborating on how pharmacy staff can play a role in identification of signs/symptoms as well and make appropriate referrals.

Line 175-176:  Are you referring to the week long training to become a certified MHFA instructor or is there a additional training to be able to use the cases created by NCPA?

Author Response

Line 8: Reviewer asked about "proper" treatment statement in Abstract.  We agree that it is vague and not well stated.  We removed the word "proper" and simply kept to "receive treatment."  We hope this makes the statement more clear. Thank you for the good suggestion.

Line 12: We deleted the word "the" between through and training.

Line 15: We removed the space between the end of the sentence and the first word of the next sentence, "Lastly."

Line 32: We removed the word "convenient" since we thought unnecessary and not critical to concept being explored.

Line 63: We did submit Figure 1 in our submission.  We hope editors will be sure to include in the final submission.

Line 99-115: We added a short piece on other roles pharmacists might be involved in. We added some text in lines 115-118.  These roles are also suggested in the MHFA model (ALGEE) that pharmacists trained in the program would presumably take on.   

Line 175-176: The opportunity to use the additional cases created by NCPA is available to those who complete the several days of NCBH training to be become certified to train MHFA.  We clarified with adding NCBH to describe the training and an additional sentence that indicates no additional training is needed for folks to use the NCPA cases.